# Recommendations and guidelines for creating scholarly biomedical journals: A scoping review

Jeremy Y. Ng[1]*, Kelly D. Cobey[2,3], Saad Ahmed[4], Valerie Chow[4], Sharleen G. Maduranayagam[4], Lucas J. Santoro[4], Lindsey Sikora[5], Ana Marusic[6], Daniel Shanahan[7], Randy Townsend[7,8], Alan Ehrlich[9,10], Alfonso Iorio[4,11], David Moher[1,2]

1 Centre for Journalology, Ottawa Methods Centre, Ottawa Hospital Research Institute, Ottawa, Ontario, Canada, 2 School of Epidemiology, Public Health and Preventive Medicine, Faculty of Medicine, University of Ottawa, Ottawa, Canada, 3 Meta-Research and Open Science Program, University of Ottawa Heart Institute, Ottawa, Canada, 4 Department of Health Research Methods, Evidence, and Impact, Faculty of Health Sciences, McMaster University, Hamilton, Ontario, Canada, 5 Health Sciences Library, University of Ottawa, Ottawa, Ontario, Canada, 6 Department of Research in Biomedicine and Health and Center for Evidence-based Medicine, University of Split School of Medicine, Split, Croatia, 7 PLOS, San Francisco, California, United States of America, 8 College of Professional Studies, George Washington University, Washington, DC, United States of America, 9 EBSCO Information Services, Ipswich, Massachusetts, United States of America, 10 Department of Family Medicine and Community Health at the UMASS Chan Medical School, Worcester, Massachusetts, United States of America, 11 Department of Medicine, McMaster University, Hamilton, Ontario, Canada

* jerng@ohri.ca

**Data Availability Statement:** All data are available from Open Science Framework: https://doi.org/10.17605/OSF.IO/XJMQW.

## Abstract

### Background

Scholarly journals play a key role in the dissemination of research findings. However, little focus is given to the process of establishing new, credible journals and the obstacles faced in achieving this. This scoping review aimed to identify and describe existing recommendations for starting a biomedical scholarly journal.

### Methods

We searched five bibliographic databases: OVID Medline + Medline in Process, Embase Classic + Embase, ERIC, APA PsycINFO, and Web of Science on January 14, 2022. A related grey literature search was conducted on March 19, 2022. Eligible sources were those published in English in any year, of any format, and that described guidance for starting a biomedical journal. Titles and abstracts of obtained sources were screened. We extracted descriptive characteristics including author name, year and country of publication, journal name, and source type, and any recommendations from the included sources discussing guidance for starting a biomedical journal. These recommendations were categorized and thematically grouped.

### Results

A total of 5626 unique sources were obtained. Thirty-three sources met our inclusion criteria. Most sources were blog posts (10/33; 30.30%), and only 10 sources were supported by

**Funding:** JYN was funded by a MITACS Accelerate Industrial Fellowship (sponsored by EBSCO Health) (ID number: IT26565). The funders had no role in study design, data collection and analysis, decision to publish, or preparation of the manuscript.

**Competing interests:** The authors declare that they have no competing interests.

**Abbreviations:** COPE, Committee on Publication Ethics; CSE, Council of Science Editors; DOAJ, Directory of Open Access Journals; DOI, Digital Object Identifier; EQUATOR, Enhancing the QUAlity and Transparency Of health Research; ERIC, Education Resources Information Center; HTML, HyperText Markup Language; ISI, Institute for Scientific Information; ISSN, International Standard Serial Number; JATS, Journal Article Tag Suite; JEP, Journal of Electronic Publishing; OASPA, Open Access Scholarly Publishers Association; OSF, Open Science Framework; PDF, Portable Document Format; PKP, Public Knowledge Project; PRESS, Peer Review of Electronic Search Strategies; PRISMA-ScR, PRISMA extension for Scoping Reviews; PRISMA, Preferred Reporting Items for Systematic Reviews and Meta-Analyses; SIR, Scientific Journal Rankings; SNIP, Source Normalized Impact per Paper; SPN, Scholarly Personal Narrative; TRU, Thompson Rivers University; VAD, Vereinigung für Afrikawissenschaften in Deutschland; WAME, World Association of Medical Editors; XML, Extensible Markup Language.

evidence. We extracted 51 unique recommendations from these 33 sources, which we thematically classified into nine themes which were: journal operations, editorial review processes, peer review processes, open access publishing, copyediting/typesetting, production, archiving/indexing/metrics, marketing/promotion, and funding.

## Conclusions

There is little formal guidance regarding how to start a scholarly journal. The development of an evidence-based guideline may help uphold scholarly publishing quality, provide insight into obstacles new journals will face, and equip novice publishers with the tools to meet best practices.

## Background

There are more than 5200 journals presently indexed in Medline [1]. This volume presents biomedical researchers with numerous journal options for submitting their research. The importance of reporting research findings completely, transparently and in credible journals is often emphasized for researchers [2]. However, less focus has been given to the role new scholarly journals have in supporting and contributing to the complete and transparent reporting of research findings, as well as the obstacles associated with this process.

In 2012, there were over 28,000 peer-reviewed journals, and this number has continued to steadily increase [3]. New journals may aim to fill niches for different or new fields of research, or other factors such as geographical location or language; additional new journals may serve as a platform to feature research of scholarly organizations, associations or societies, [3, 4]. Therefore, in order for a new journal to best support the community it serves, the publisher (e.g., publishing group, university/academic institute, association, etc.) needs to first clearly identify and articulate the aim and scope of the new journal. This statement of aims and scope should outline a brief introduction to the journal's overall objectives, the unmet need that it aims to fill, which subjects it covers, and the types of articles it publishes [5, 6]. This helps provide context to underpin a series of important decisions that need to be made for the journal to achieve its mission; including, but not limited to, the journal's content type, peer review policy, publication schedule, author guidelines, distribution/licensing/publishing agreements and the editorial/submission policies [7]. The decisions made with respect to each of these issues need to reflect the specific behaviours and needs of the target community, for the journal to achieve its objectives.

It is worth briefly mentioning that careful and ethical consideration in evaluating the need for a new scholarly journal would prevent launching predatory journals. Predatory journals are mostly driven by achieving profit, thus accepting manuscript submissions not for their quality or innovative content, but solely to receive financial profits from author publication fees [8]. Predatory journals often present false or misleading information about their peer review practices, may have an unverified editorial board, aggressively solicitate researchers for submissions, and lack transparency [8]. For publishers of new journals, it is thus important that they are aware of the appropriate publishing practices such that they are not labelled as predatory. Building credibility and trust is a major obstacle in starting nearly all new scholarly journals [5, 9].

Thus, in addition to these community-specific considerations, adhering to established best practices, standards, and emerging norms are likely important credibility steps when starting a

new journal. The editors and editorial boards should be selected to support the scope and mission of the journal, but they must also be educated on the various existing publishing standards to ensure a scholarly journal's publishing processes are updated and compliant with best publication scholarship practices [5]. For example, a consensus on best publishing practices developed via collaboration between the Committee on Publication Ethics (COPE), the Directory of Open Access Journals (DOAJ), the Open Access Scholarly Publishers Association (OASPA), and the World Association of Medical Editors (WAME) provides a comprehensive understanding of guiding principles in the journal publishing process [10].

Given the wide range of considerations and activities involved in successfully starting a new journal, formal guidance on how to navigate journal creation would be very valuable to large and small (prospective) publishers alike. Such guidance could support equity through providing formal instructions on best practices for the range of stakeholders across varied jurisdictions and resource capacities for starting new journals. This type of guidance could also act to safeguard the quality of scholarly publishing by providing minimum criteria required to launch a journal. Guidance in this filed was issued already by COPE, DOAJ, OASPA, and WAME [10], but there is not (yet) a broad consensus in this space. The purpose of this review is to identify and categorize all recommendations that exist both within and outside of the above organizations, something that prior to this review was unknown. The aim of this study was to conduct a scoping review to identify and map existing guidance documents for starting biomedical journals. We focused on biomedicine as the community behaviours and drivers that a new journal would need to reflect could be expected to be broadly similar within this field.

## Methods

### Approach

We developed a protocol before initiating this study. The protocol was registered and included on the Open Science Framework (OSF) [11] before data collection and analysis. Study materials and data are shared on the OSF: https://doi.org/10.17605/OSF.IO/XJMQW. We reported the completed scoping review using the PRISMA-ScR for scoping reviews statement [12].

### Step 1: Identifying the research question

In the present scoping review, we sought to answer the following question: What recommendations exist for starting a scholarly biomedical journal? We acknowledge that a plethora of business- and research-related decisions need to be made to establish a scholarly journal. In this review, we were interested in collecting recommendations pertaining to the latter. For example, once the decision has already been made by journal founders to establish a journal in a particular area of biomedicine, in this review, we captured and mapped guidance documents on what the key decisions are that should proceed this. The five steps of our scoping review approach are: 1) identifying the research question, 2) identifying relevant studies, 3) selecting the studies, 4) charting the data, and 5) collating, summarizing, and reporting the results [13, 14].

### Step 2: Finding relevant studies

**Database search.**   The following databases were searched by a health sciences librarian (LS): Medline and Medline in Process via Ovid (1946 –January 13, 2022), Embase Classic + Embase via Ovid (1947 –January 13, 2022), APA PsycINFO via Ovid (1806 –Week 1, January 2022), ERIC via Ovid (1965 to May 2021), and Web of Science Core Collection, Science

Citation Index Expanded [SCI-EXPANDED, 1900 –present] (searched January 14, 2022). A search strategy was developed in Medline by an information specialist (LS), and then translated into the other databases, as appropriate (see **Table 1**). All search strategies were peer reviewed using the PRESS tool [15] by another academic librarian (N. Langlois). All databases were searched from their dates of inception to January 14, 2022. There were no publication restrictions. All references were entered into an Endnote file for processing, and then DistillerSR for deduplication and screening [16, 17].

**Grey literature search.** We also conducted a grey literature search to capture guidance documents published outside of the standard peer-reviewed literature (e.g., policy papers or blogs released by research, editorial or publishing organizations). Organizations which may be sources of eligible grey literature (e.g., relevant guidance provided on webpages of publisher websites, and publisher organizations/association websites such as COPE and DOAJ) were

**Table 1. Bibliographic database search strategies for articles providing guidance for the creation of new scholarly journals, executed January 14, 2022.**

| Medline |
| --- |
| 1. (Periodicals as Topic/ or (periodical or periodicals or journal or journals).ti,ab.) adj1 (successful or start* or create or creating or created).ti,ab. |
| 2. (Periodicals as Topic/ or (periodical or periodicals or journal or journals).ti,ab.) adj2 (guidance or advice or advise or suggest* or recommend*).ti,ab. |
| 3. 1 or 2 |
| **Embase** |
| 1. (medical literature/ or (periodical or periodicals or journal or journals).ti,ab.) adj1 (successful or start* or create or creating or created).ti. |
| 2. (medical literature/ or (periodical or periodicals or journal or journals).ti,ab.) adj2 (guidance or advice or advise or suggest* or recommend*).ti. |
| 3. 1 or 2 |
| **ERIC** |
| 1. ((periodical or periodicals or journal or journals) adj2 (guidance or advice or advise or suggest* or recommend*)).ti,ab. |
| 2. ((periodical* or journal or journals) adj1 (successful or start* or creat*)).ti,ab. |
| 3. 1 or 2 |
| **APA PsycINFO** |
| 1. ((periodical or periodicals or journal or journals) adj2 (guidance or advice or advise or suggest* or recommend*)).ti,ab. |
| 2. ((periodical* or journal or journals) adj1 (successful or start* or creat*)).ti,ab. |
| 3. 1 or 2 |
| **Web of Science** |
| Query #1 |
| TI = ((journal or journal or periodical or periodicals) NEAR/1 (successful or start* or create or creating or created)) |
| Query #3 |
| (AB = ((journal or journal or periodical or periodicals) NEAR/1 (successful or start* or create or creating or created))) |
| Query #4 |
| TI = ((periodical or periodicals or journal or journals) NEAR/2 (guidance or advice or advise or suggest* or recommend*)) |
| Query #5 |
| AB = ((periodical or periodicals or journal or journals) NEAR/2 (guidance or advice or advise or suggest* or recommend*)) |
| Query #6 |
| #5 OR #4 OR #3 OR #1 |

sourced and prepared into a list by JYN and then reviewed by the research team. A complete list is found in **Table 2**. Each organization's website was searched using the following search term: "starting a new scholarly journal"; the search bar was used if it was present on a given website, however, if there was none, we searched the site via Google (i.e., "site.website.com"). Separate to these websites, we also conducted general searches on Google.com and YouTube. com using the same search terms, and the first 100 results returned from the search from both websites was reviewed for eligibility, so that items outside of our generated list of organizations were also captured. Lastly, once all eligible items were identified, we reviewed each item's reference list, and sources that have cited each item (if possible), to identify any additional eligible items.

### Step 3: Selecting the studies

Documents that characterised and/or described guidance for creating a new biomedical scholarly journal were included in the study population, regardless of whether the documents were evidence-based. All study designs and narrative documents pertaining to the field of biomedicine were included. We placed no limitations on geographic region or year the document was produced. All non-English content was excluded for feasibility.

### Step 4: Charting the data

Prior to data extraction, data extraction forms were piloted (see **Table 3**). Five authors (JYN, VC, LJS, SA, SGM) participated in pilot data extractions. Three authors (JYN, VC, LJS) conducted a pilot data extraction for bibliographic sources, while three authors (JYN, SGM, SA) conducted a pilot extraction for grey literature sources. Each author independently conducted data extractions of the first four sources using the developed data extraction form. Once completed, the authors of each respective team met to compare results, resolved conflicts, and formulated a revised data extraction form. This revised form was then used for all following extractions. The remaining bibliographic sources were divided equally between authors VC and LJS, while JYN reviewed all bibliographic sources. The remaining grey literature sources were divided equally between authors SA and SGM., with author JYN resolving conflicts. First, we reviewed the titles and abstracts of articles obtained using our screening question both independently and in duplicate. We then checked that full-text articles of those that passed title and abstract screening met the inclusion criteria, then extracted the following information: corresponding author's name/organizations name, country, year of publication (we selected the most recent date stated), study design (as determined by the reviewers), and journal name. Additionally, we also categorized recommendations provided from each article pertaining to the creation of a new journal.

### Step 5: Collating, summarizing, and reporting the results

Quantitative (i.e., frequencies) and qualitative (i.e., thematic analysis) methods were used to analyse our data. A list of recommendations for creating a new journal resulting from the data extraction was generated in duplicate by SA and SGM. Each of the recommendations extracted from the included sources were initially grouped into predetermined categories loosely based on Cobey et al. [18]. If a recommendation did not fit into an existing category, a new category was created. Each recommendation was coded independently by three authors (JYN, SA, SGM), who upon completion met to finalize all coded recommendations. Included sources were also categorized by evidence dimension (i.e., whether recommendations were informed by evidence/research or by expert opinion/expertise). Following this, descriptive data was collected including author name, year and country of publication, journal name, and source type.

**Table 2. List of grey literature sources.**

| Organization Name | Website |
| --- | --- |
| CLOCKSS | https://clockss.org/ |
| LOCKSS | https://www.lockss.org/ |
| Portico | https://www.portico.org/ |
| PubMed Central | https://www.ncbi.nlm.nih.gov/pmc/ |
| SCOPUS | https://www.scopus.com/ |
| OVID | https://www.ovid.com/ |
| ResearchGate | https://www.researchgate.net/ |
| Academia.edu | https://www.academia.edu/ |
| Web of Science | https://www.webofknowledge.com/ |
| Semantic Scholar | https://www.semanticscholar.org/ |
| EBSCO | https://www.ebsco.com/ |
| WorldCat | https://www.worldcat.org/ |
| PsycInfo | https://www.apa.org/pubs/databases/psycinfo |
| Citeseer | https://citeseerx.ist.psu.edu/ |
| ScienceOpen | https://www.scienceopen.com/ |
| Europe PMC | https://europepmc.org/ |
| Scholar's Portal | https://scholarsportal.info/ |
| ProQuest | https://search.proquest.com/ |
| Google Scholar | https://scholar.google.ca/ |
| ResearchSquare | https://www.researchsquare.com/ |
| MedRxiv | https://www.medrxiv.org/ |
| Open Science Foundation Preprints | https://osf.io/preprints/ |
| Eprints | https://www.eprints.org/uk/ |
| Nature | https://www.nature.com/ |
| Wiley | https://www.wiley.com/en-ca |
| Elsevier | https://www.elsevier.ca/ca/ |
| Wolters Kluwer | https://www.wolterskluwer.com/en-ca |
| Clarivate | https://clarivate.com/ |
| Routledge/Taylor and Francis | https://taylorandfrancis.com/ |
| Sage Publications | https://us.sagepub.com/en-us/nam/home |
| PLoS | https://plos.org/ |
| PeerJ | https://peerj.com/ |
| F1000 Research | https://f1000research.com/ |
| BioMedCentral | https://www.biomedcentral.com/ |
| Hindawi | https://www.hindawi.com/ |
| Frontiers | https://www.frontiersin.org/ |
| JSTOR | https://www.jstor.org/ |
| Karger | https://www.karger.com/ |
| DeGruyter | https://www.degruyter.com/ |
| IngentaConnect | https://www.ingentaconnect.com/ |
| World Scientific Publishing | https://www.worldscientific.com/ |
| Palgrave MacMillan | https://www.palgrave.com/us |
| MDPI | https://www.mdpi.com/ |
| Mary Ann Liebert | https://www.liebertpub.com/ |
| Begell House | https://www.begellhouse.com/ |
| Lancet | https://www.thelancet.com/ |
| Cambridge University Press | https://www.cambridge.org/ |

(*Continued*)

**Table 2.** (Continued)

| Organization Name | Website |
| --- | --- |
| Oxford University Press | https://global.oup.com/?cc=ca |
| Dove Medical Press | https://www.dovepress.com/ |
| Canadian Science Publishing | https://cdnsciencepub.com/ |
| eLife | https://elifesciences.org/ |
| International Association of Scientific, Technical, and Medical Publishers | https://www.stm-assoc.org/ |
| Committee on Publication Ethics | https://publicationethics.org/ |
| CrossRef | https://www.crossref.org/ |
| International Committee of Medical Journal Editors | http://www.icmje.org/ |
| World Association of Medical Editors | http://www.wame.org/ |
| European Association of Science Editors | https://ease.org.uk/ |
| Council of Science Editors | https://www.councilscienceeditors.org/ |
| Open Access Scholarly Publishers Association | https://oaspa.org/ |
| Open Knowledge Foundation | https://okfn.org/ |
| Creative Commons | https://creativecommons.org/ |
| Open Society Foundations | https://www.opensocietyfoundations.org/ |
| Public Knowledge Project | https://pkp.sfu.ca/ |
| Scholarly Publishing and Academic Resources Coalition | https://sparcopen.org/ |
| International Network for Advancing Science and Policy | https://www.inasp.info/ |
| Our Research | https://ourresearch.org/ |
| Canadian Association of Research Libraries | https://www.carl-abrc.ca/advancing-research/scholarly-communication/open-access/?cn-reloaded=1 |
| Canadian Research Knowledge Network | https://www.crkn-rcdr.ca/en |
| Association of Canadian University Presses | http://acup-apuc.ca/ |
| Mendeley | https://www.mendeley.com/ |
| Kopernio | https://kopernio.com/ |
| Altmetric | https://www.altmetric.com/ |
| UNESCO | https://en.unesco.org/science-sustainable-future/open-science/partnership |
| World Health Organization | https://www.who.int/about/policies/publishing/open-access |
| Scholastica | https://scholasticahq.com/ |
| Directory of Open Access Journals | https://doaj.org/ |

The goal was to identify and thematically group characteristics of recommendations made for the creation of a new scholarly journal, so we did not formally assess the methodological quality of the studies/documents from which the characteristics were extracted. This process resulted in 51 unique recommendations across thirteen categories which included journal operations, editorial review processes, peer review processes, open access publishing, copyediting, production, indexing, archiving, patient engagement, journal metrics, obtaining peer reviewers, and article processing charges. After this initial categorization, we grouped similar categories into broader themes using a deductive thematic analysis approach. This was completed in duplicate by SA and SGM, with JYN resolving any discrepancies. This step resulted in nine themes which included journal operations, editorial review processes, peer review processes, open access publishing, copyediting/typesetting, production, archiving/indexing/metrics, marketing/promotion, and funding.

**Table 3. Data extraction forms.**

| |
|---|
| **Level 1.** Title and abstract screening form |
| 1. Is this article written in English? |
| Yes/No |
| 2. Does this article discuss guidance for the creation of new journals? |
| Yes/Unclear, or No |
| **Level 2.** Full text article screening |
| 1. Is there any reason this article should be excluded? |
| a. Yes–Not in English |
| b. Yes–Does not provide guidance to scholarly publishers for the creation of new journals |
| c. Yes–No full text |
| d. Yes–other please specify |
| e. No |
| **Level 3: Extraction form** |
| 1. What year was the work published? (choose latest appearing date, if no date is listed indicate 'unknown'): |
| 2. What journal was the work published in? (Use capitals for each word, and abbreviations if they are common): e.g., "Nature" "BMC Medicine" "JAMA" "BMJ": |
| 3. What is the name of the corresponding author? If the corresponding author is not specified use first author's details. (write as first last: e.g., David Moher; Kelly D. Cobey; K.D. Cobey). If no author names are provided, please provide the organization name (e.g., International Committee of Medical Journal Editors). If you can't discern this, please write "not reported". |
| 4. What country is listed in the primary affiliation of the corresponding author? If corresponding author not specified use first author. If city and university are listed, but not country, state country if you can easily infer this, but don't search affiliations online. If multiple affiliations are listed which have different countries, extract the country from the first stated affiliation. If no author names are provided, please provide the affiliation of the organization's headquarters. If you can't discern this, please write "not reported". |
| 5. What type of work is described? Assess this based on your evaluation of study design, not based on what the labelling of the article states. |
| a. Commentary/Viewpoint/Editorial/Letter |
| b. Systematic Review (must describe databases searched >1, search dates must be specified, eligibility criteria described) |
| c. Narrative Review |
| d. Case report/Case Series |
| e. Observational study |
| f. RCT |
| g. Other, please specify |
| 6. Please data extract all recommendations made by the work regarding the creation of a new journal. For the purpose of this study, a recommendation is defined as a course of action suggested as suitable or appropriate for an scholarly publisher to take when creating a new journal. |
| 7. Please categorize the extracted recommendations using the categories listed below: |
| a. Journal operations |
| b. Editorial review processes |
| c. Peer review processes |
| d. Open access publishing |
| e. Copyediting |
| f. Production |
| g. Indexing |
| h. Archiving |
| i. Patient engagement |
| k. Journal metrics |
| l. Obtaining peer reviewers |
| m. Article processing charges |

*(Continued)*

**Table 3.** (Continued)

| |
|---|
| n. Other, please specify |
| 8. What stakeholders were involved in the development of these recommendations (e.g., industry, researchers, societies, etc.)? |
| a. Scholarly publishers |
| b. Publishing organization members |
| c. Journal editors |
| d. Researchers |
| e. Other, please specify |
| 9. Were the criteria for selecting the evidence used to formulate these recommendations clearly described? |
| Yes–systematic search underpins recommendations |
| Yes–empirical work cited to underpin recommendations |
| Yes–formal consensus process used (eg. Delphi) to create recommendations |
| Yes–other, please specify |
| No |
| 10. Were systematic methods used to search for evidence which informed these recommendations? |
| 11. Was any information provided with respect to the barriers and/or facilitators to the implementation and uptake of these recommendations? If yes, please list them here. |
| 12. Were any potential resource implications of applying the recommendations considered? |
| Yes |
| No |
| 13. Comments: |

## Results

### Search results

Searches generated a total of 5627 records, of which 5351 of them were titles/abstracts of bibliographic sources and 276 records were grey literature sources. Following title/abstract screening of the 5351 bibliographic sources, 4624 titles/abstracts were excluded, leaving 727 bibliographic full texts to be assessed. In combination with the 276 grey literature sources, a total of 1003 full text sources were screened in this review. A total of 966 sources were excluded because they did not provide guidance to scholarly publishers for the creation of new journals (n = 919), were published in a language other than English (n = 45) or were irretrievable (n = 2). After full-text screening, 37 sources were deemed eligible. Of these 37 sources, 4 were duplicates, resulting in a total of 33 eligible sources that were included in this review. A PRISMA diagram detailing this process is depicted in **Fig 1**.

### Eligible source characteristics

The 33 sources that met the inclusion criteria were published between 1998 and 2022 (See **Table 4**). These sources originated from the United States (n = 17), the United Kingdom (n = 5), Canada (n = 3), India (n = 2), Australia (n = 1), France (n = 1), Germany (n = 1), Pakistan (n = 1), the Philippines (n = 1), and Saudi Arabia (n = 1). While the majority of sources were blog posts (n = 10), there were also informational guides (n = 8), research articles (n = 6), YouTube videos (n = 3), organizational reports (n = 2), a news article (n = 1), an editorial (n = 1), and a transcript of a presentation (n = 1). The number of sources providing guidance for the creation of new scholarly journals increased from 1998 to 2022.

A total of 433 recommendations pertaining to scholarly journal creation were extracted. Of the 33 sources providing guidance for scholarly journal creation, only 10 were informed by evidence. Of these 10 sources: 8 cited at least a proportion of evidence-based research to

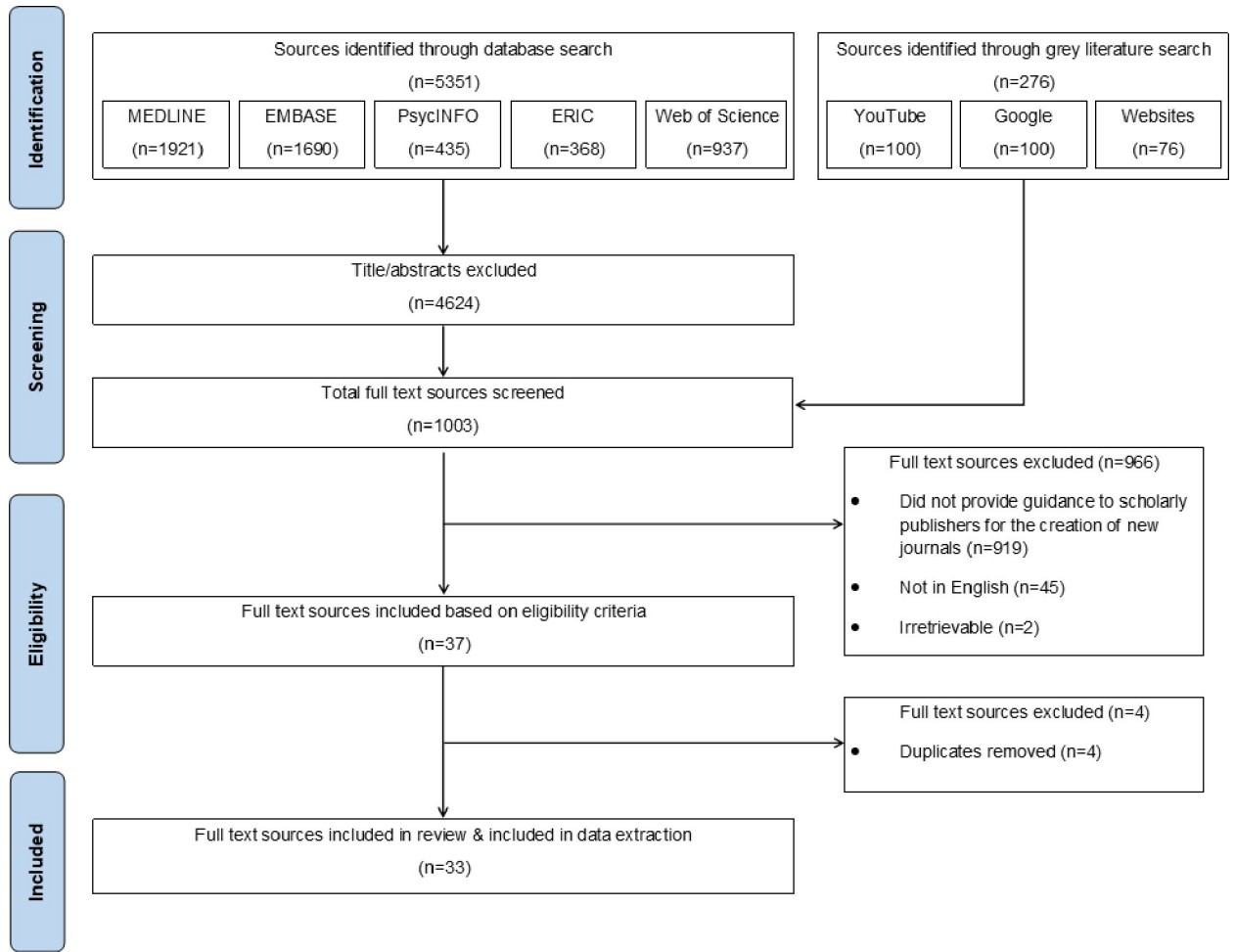

**Fig 1. PRISMA flow diagram summarizing search process and source selection.**

support their recommendations; one source conducted a formal investigation to inform their recommendations; and one source gathered information on journal creation through a consensus approach with their peers but did not publish their findings. The majority of sources used expert opinion and/or personal experience as either a journal publisher or journal editor (23/33) to inform their recommendations. Further details associated with all eligible sources referenced in this review can be found in **Table 5**.

## Mapping the data into emergent themes

The extracted recommendations were categorized into nine themes [18–24] (See **Table 6** for themes and definitions), and a recommendation addressing more than one theme was coded multiple times (See **Fig 2**). All sources made multiple recommendations across different themes, with the number of extracted recommendations ranging from 3 to 32 (**Table 5**). Of the 33 sources, the number of sources that made at least one recommendation in each theme are as follows: journal operations (n = 31), editorial review processes (n = 30), peer review processes (n = 20), open access publishing (n = 19), copyediting/typesetting (n = 14), production (n = 21), indexing/archiving/metrics (n = 13), marketing/promotion (n = 15), funding models (n = 24). (See **Table 7**).

**Table 4. Characteristics of all included sources mentioning recommendations for new scholarly journal creation and characteristics of included evidence-based sources.**

| | Sources providing guidance for creating new scholarly journals (n = 33) | Evidence-based sources included in scoping review (n = 10) |
|---|---|---|
| **Nationality of corresponding authors** | USA: 17 | USA: 5 |
| | UK: 5 | France: 1 |
| | Canada: 3 | India: 1 |
| | India: 2 | Pakistan: 1 |
| | Australia: 1 | Saudi Arabia: 1 |
| | France: 1 | UK: 1 |
| | Germany: 1 | |
| | Pakistan: 1 | |
| | Philippines: 1 | |
| | Saudi Arabia: 1 | |
| **Publication year of sources** | 1998: 1 | 2004: 1 |
| | 2004: 1 | 2005: 1 |
| | 2005: 1 | 2008: 1 |
| | 2008: 1 | 2009: 1 |
| | 2009: 1 | 2012: 1 |
| | 2012: 1 | 2016: 1 |
| | 2014: 2 | 2017: 1 |
| | 2015: 2 | 2018: 2 |
| | 2016: 3 | 2020: 1 |
| | 2017: 2 | |
| | 2018: 5 | |
| | 2019: 5 | |
| | 2020: 1 | |
| | 2021: 3 | |
| | 2022: 2 | |
| | Not reported: 2 | |
| **Source design** | Blog Post: 10 | Research Article: 5 |
| | Guide/Booklet: 8 | Guide/Booklet: 3 |
| | Research Article: 6 | Editorial: 1 |
| | YouTube Video: 3 | Organizational Report: 1 |
| | Organizational Report: 2 | |
| | News Article: 1 | |
| | Editorial: 1 | |
| | Forum: 1 | |
| | Transcript of Presentation: 1 | |

From the 33 sources, a total of 51 unique recommendations were extracted and categorized into nine themes (See **Table 7**):

**Journal operations.** The most common recommendations in the journal operations theme were to identify the gap/niche the new journal will fill (n = 26 sources), build a website for the new journal (n = 15 sources), and determine how often the new journal will publish (n = 15 sources). Technical recommendations in this theme were also common, with 14 sources recommending utilizing manuscript-management software/system (e.g., DPubS, ePublishing Toolkit, OpenACS, Open Journal Systems) to keep track of submissions, papers out for review, those accepted/rejected for publication), and 13 sources each recommending

**Table 5. General characteristics of included evidence-based sources (n = 33).**

| Web Source/Source Title | Source DOI/PMID/ URL | Year of Publication | Corresponding Author/ Organization | Country of Corresponding Author | Source Type | Number of Themes of Extracted Recommendations | Number of Extracted Recommendations |
|---|---|---|---|---|---|---|---|
| The University of Kansas Libraries | https://guides.lib.ku.edu/journal_editors/launching | 2022 | University of Kansas | USA | Guide | 6 | 11 |
| How to Start a Scholarly Open Access Journal With OJS | https://www.youtube.com/watch?v=r14n5BAoHeY | 2022 | WavesThink | Philippines | YouTube Video | 3 | 3 |
| Typeset | https://typeset.io/resources/how-to-start-an-open-access-journal/ | 2021 | Shanu Kumar | USA | Blog | 7 | 18 |
| Scholastica | https://blog.scholasticahq.com/post/how-to-start-flip-open-access-academic-journal/ | 2021 | Scholastica | USA | Blog | 8 | 24 |
| Thompson Rivers University (TRU) Libraries | https://libguides.tru.ca/ojs/content | 2021 | TRU Libraries | Canada | Guide | 5 | 13 |
| Creation of a Peer-Reviewed Journal as a Catalyst for Innovative Nursing Research and Enhancing Evidence-Based Nursing Practice | https://doi.org/10.1097/nna.0000000000000907 | 2020 | Elizabeth B. Card | USA | Research Article | 4 | 11 |
| Launching a Journal: Part 1: Research & Development | https://www.youtube.com/watch?v=nAgsvATb5e0 | 2019 | John Bond | USA | YouTube Video | 4 | 7 |
| Columbia University | https://www.cuimc.columbia.edu/news/thinking-about-starting-your-own-academic-journal | 2019 | Columbia University | USA | News Article | 7 | 8 |
| Judith Johnson | https://judithjohnsonphd.com/2019/04/08/how-to-start-a-journal-and-beat-the-academic-publishing-racket/ | 2019 | Judith Johnson | United Kingdom | Blog | 7 | 15 |
| Scholastica | https://blog.scholasticahq.com/post/road-to-oa-starting-flipping-journal/ | 2019 | Scholastica | USA | Blog | 6 | 16 |
| Western Libraries | https://www.lib.uwo.ca/files/scholarship/Starting%20a%20New%20Journal%20-%20Does%20it%20make%20sense.pdf | 2019 | David J. Solomon | Canada | Guide | 6 | 10 |
| Editage | https://www.editage.com/insights/how-do-i-create-my-own-peer-reviewed-open-access-journal | 2018 | Editage | India | Forum | 3 | 6 |
| Digital Commons | https://digitalcommons.uncfsu.edu/jri/vol3/iss3/13 | 2018 | Sydney Freeman Jr | USA | Research Article | 8 | 23 |
| How to Start or Flip an Open Access Journal: Publishers and editors share their stories | https://www.youtube.com/watch?v=Z6NxgwH0FD0 | 2018 | Scholastica | USA | YouTube Video | 7 | 19 |

*(Continued)*

**Table 5.** (Continued)

| Web Source/Source Title | Source DOI/PMID/ URL | Year of Publication | Corresponding Author/ Organization | Country of Corresponding Author | Source Type | Number of Themes of Extracted Recommendations | Number of Extracted Recommendations |
|---|---|---|---|---|---|---|---|
| International Network for Advancing Science and Policy | https://www.inasp.info/sites/default/files/2018-04/INASP%20-%20Editors%20Toolkit%20-%20DIGITAL.pdf | 2018 | Pippa Smart | United Kingdom | Guide | 9 | 36 |
| Vereinigung für Afrikawissenschaften in Deutschland (VAD) | https://www.vad-ev.de/en/checklist-for-journal/ | 2018 | VAD | Germany | Blog | 5 | 10 |
| How to run a successful journal | https://dx.doi.org/10.12669%2Fpjms.336.14097 | 2017 | Shaukat Ali Jawaid | Pakistan | Research Article | 7 | 20 |
| Journal of Electronic Publishing (JEP) | https://quod.lib.umich.edu/j/jep/3336451.0020.209?view=text;rgn=main | 2017 | Richard L. Saunders | USA | Presentation | 7 | 15 |
| Scholarly Kitchen | https://scholarlykitchen.sspnet.org/2016/08/04/nuts-and-bolts-the-super-long-list-of-things-to-do-when-starting-a-new-journal/ | 2016 | Angela Cochran | USA | Blog | 5 | 15 |
| Martin Paul Eve | https://eve.gd/2012/07/10/starting-an-open-access-journal-a-step-by-step-guide-part-1/ | 2016 | Martin Paul Eve | United Kingdom | Blog | 6 | 12 |
| The Indian Anaesthetists' Forum | https://www.theiaforum.org/article.asp?issn=2589-7934;year=2016;volume=17;issue=1;spage=3;epage=5;aulast=Trikha | 2016 | Anjan Trikha | India | Editorial | 5 | 8 |
| Elgar Blog | https://elgar.blog/2015/06/04/starting-a-new-journal-by-ben-booth/ | 2015 | Ben Booth | United Kingdom | Blog | 3 | 3 |
| Council of Science Editors (CSE) | http://www.councilscienceeditors.org/wp-content/uploads/v38n3_4WebPdf.pdf | 2015 | CSE | USA | Report | 6 | 10 |
| BioMedCentral | https://blogs.biomedcentral.com/bmcblog/2014/06/17/what-does-it-take-to-run-your-own-journal/ | 2014 | Zaheer-Ud-Din Babar | Australia | Blog | 4 | 6 |
| Public Knowledge Project (PKP) | https://pkp.sfu.ca/pkp-software-documentation/new-journal-workplan/ | 2014 | PKP | USA | Guide | 6 | 10 |
| UNESCO | https://en.unesco.org/open-access/sites/open-access/files/215863e.pdf | 2012 | Alma Swan | France | Guide | 2 | 4 |
| The Web Journal of Mass Communication Research | https://wjmcr.info/2009/03/01/a-bakers-dozen-of-issues-facing-online-academic-journal-start-ups/ | 2009 | Thomas Gould | USA | Research Article | 7 | 18 |
| Public Knowledge Project | https://pkp.sfu.ca/files/AfricaNewJournal.pdf | 2008 | Kevin Stranack | USA | Guide | 9 | 32 |

(*Continued*)

**Table 5.** (Continued)

| Web Source/Source Title | Source DOI/PMID/ URL | Year of Publication | Corresponding Author/ Organization | Country of Corresponding Author | Source Type | Number of Themes of Extracted Recommendations | Number of Extracted Recommendations |
|---|---|---|---|---|---|---|---|
| Public Knowledge Project | https://pkp.sfu.ca/files/OJS_Project_Report_Shapiro.pdf | 2005 | Lorna Shapiro | USA | Report | 7 | 12 |
| Editing and publishing of a medical journal: success of an unconventional workflow | PMID: 14968185 | 2004 | Sajjeev X. Antony | Saudi Arabia | Research Article | 5 | 10 |
| Editor Creates Journal | https://doi.org/10.1192/S0007125000150901 | 1998 | John Lewis Crammer | United Kingdom | Research Article | 2 | 5 |
| Wendy Belcher | https://wendybelcher.com/writing-advice/manage-peer-reviewed-journal/ | Unknown | Wendy Belcher | USA | Blog | 7 | 18 |
| University of Toronto Libraries | https://jps.library.utoronto.ca/index.php/pubguide/starting | Unknown | University of Toronto Libraries | Canada | Guide | 4 | 4 |

Abbreviations: Legend: DOI = Digital Object Identifier, PMID = PubMed IDentifier, URL = Uniform Resource Locator, TRU Libraries = Thompson Rivers University Libraries, VAD = Vereinigung für Afrikawissenschaften in Deutschland, JEP = Journal of Electronic Publishing, CSE = Council of Science Editors, PKP = Public Knowledge Project, UNESCO = United Nations Educational, Scientific and Cultural Organization

applying for an International Standard Serial Number (ISSN) and registering a Digital Object Identifier (DOI) for each article in the new journal. The journal operations theme also saw the highest number of unique recommendations across all nine themes (n = 11 recommendations).

**Editorial review processes.** The most common recommendation in the editorial review processes, as well as the most common recommendation across any of the nine themes was to set up an editorial board (n = 30 sources). Other common recommendations were to create a

**Table 6. Themes and definitions used to code categories of recommendations for creating a new scholarly journal.**

| Theme | Definition |
|---|---|
| Journal operations | Features related to how a new journal can conduct its business operations [18] |
| Editorial review processes | Any aspect of the internal or external review of submitted articles and decisions on what to publish [18] |
| Peer review processes | Any aspect of the peer review system used to critically assess scholarly material before it is published [19] |
| Open access publishing | Features related to making a new journal's publishing model digital, online, free of charge, and/or free of copyright and licensing restrictions [20] |
| Copyediting/ Typesetting | Making use of copyeditors and typesetters to ensure correctness, accuracy, consistency, and completeness prior to publishing scholarly material [21] |
| Production | Features related to how a new journal can produce and disseminate scholarly material [18] |
| Indexing/Archiving/ Metrics | Information on how a new journal can make use of indexing tools, archiving tools, and quantitative metric tools [18, 22] |
| Marketing/Promotion | The activity, set of institutions, and processes for creating, communicating, delivering, and exchanging offerings that have value for readers, partners, and society at large [23] |
| Funding models | Is a methodical and institutionalized approach to building a reliable revenue base that will support an organization's core programs and services [24] |

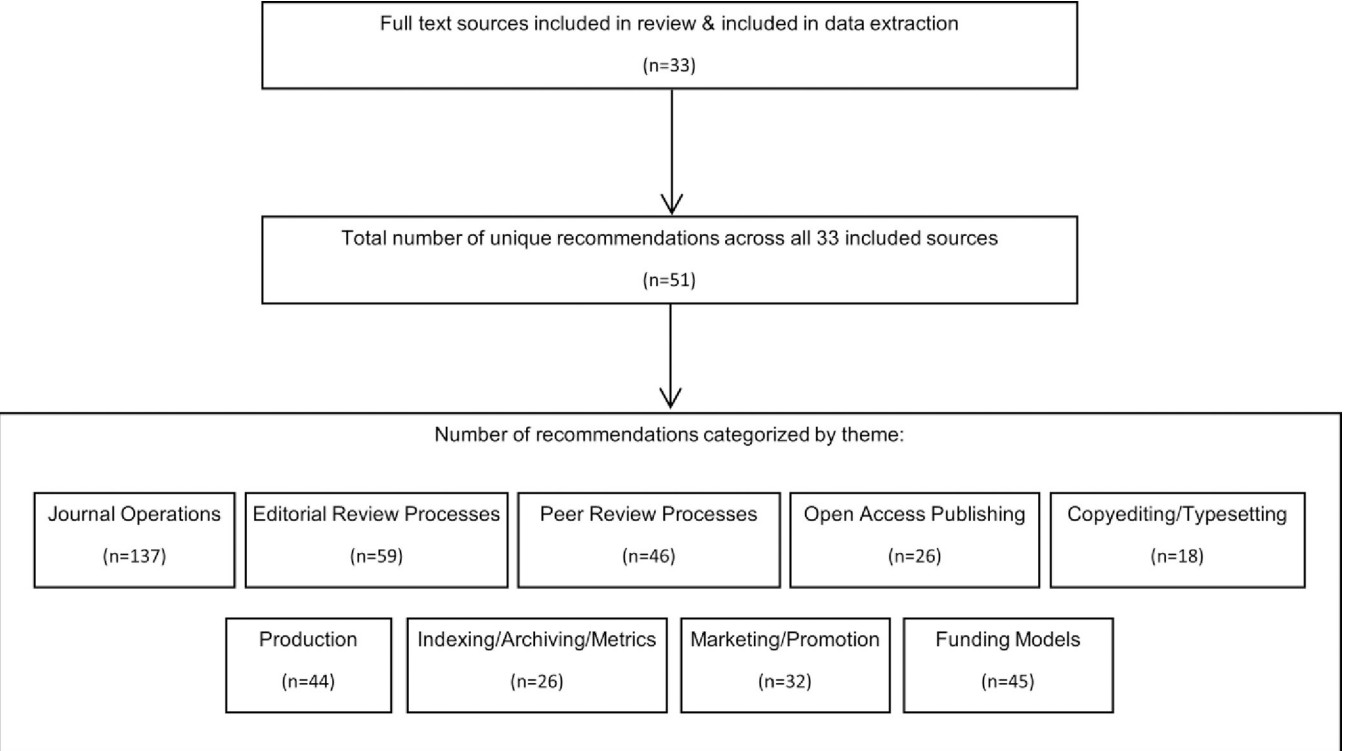

**Fig 2. Flow diagram summarizing the number of recommendations extracted from all included sources, the number of unique recommendations found, and the categorization of recommendations by theme.** *The total number of unique recommendations across all 33 sources does not equal to the total number of individual recommendations in the bottom box as recommendations can overlap across categories and individual recommendations regarding the same idea/statement were grouped to produce a count of unique recommendations.

detailed editorial meeting structure (e.g., involving reports of individual accountability, discussion of matters requiring decisions) (n = 10 sources), devise standards to evaluate articles for inclusion (n = 8 sources) and detail editor term lengths and the process for replacement/reappointment of editorial board members (n = 5 sources).

**Peer review processes.** Common recommendations in the peer review processes theme were to create a comprehensive protocol for peer review (e.g., detailing how many peer-reviewers are to be assigned per article, the specific criteria for review, the process of solving discrepancies among reviewers) (n = 13 sources), contacting potential reviewers who are working closely in the field of the journal (n = 12 sources) and setting a timeline for peer reviewer feedback and comment completion (n = 9 sources). Notably, only one source recommended to provide reviewers with monetary compensation/honorarium as an incentive for review. No sources made mention of peer reviewer training.

**Open access publishing.** In the open access publishing theme, thirteen sources each recommended to utilize specific software to set up and manage open access journals such as Open Journal Systems and determine whether the new journal will be open-access, available only by subscription, or exist as a hybrid model.

**Copywriting/Typesetting.** Recommendations in the copywriting/typesetting theme were to determine whether copyediting and typesetting will be outsourced or completed in-house (n = 12 sources) and to utilize specific software such as QuarkXpress, Adobe FrameMaker, and Adobe InDesign to copyedit and typeset articles (n = 5 sources).

**Production.** The three most common recommendations in the production theme were to determine the format of the new journal (i.e., print, online or both) (n = 18 sources),

**Table 7. All extracted recommendations regarding new scholarly journal creation from all included sources, categorized by source reference, theme, the unique recommendation found, and the total number of sources suggesting the unique recommendation.**

| Theme | Derived Unique Recommendation | Number of Sources Suggesting Recommendation | Source Identifier |
|---|---|---|---|
| **Journal Operations** | 1. Suggests to identify the gap/need/niche a new journal will fill. | 26 | Johnson (2019), University of Kansas (2022), University of Toronto (Unknown), Columbia University (2019), Babar (2014), Kumar (2021), Editage (2018), Cochran (2016), Saunders (2017), Eve (2016), Freeman Jr (2018), Gould (2009), Booth (2015), Belcher (Unknown), Trikha (2016), TRU Libraries (2021), Solomon (2019), Bond (2019), Scholastica (2018), CSE (2015), Stranack (2008), Smart (2018), Scholastica (2021), Scholastica (2019), Card (2020), Crammer (1998) |
| | 2. Suggests to build a website for your journal | 15 | Johnson (2019), Kumar (2021), Editage (2018), Cochran (2016), Saunders (2017), VAD (2018), Eve (2016), Belcher (Unknown), Scholastica (2018), Stranack (2008), Shapiro (2005), PKP (2014), Smart (2018), Scholastica (2021), Jawaid (2017) |
| | 3. Suggests specific steps to build a website (purchase a domain name, find a web-hosting platform, etc.) | 5 | Johnson (2019), Cochran (2016), Stranack (2008), PKP (2014), Smart (2018) |
| | 4. Suggests the use of manuscript-management software/ system to keep track of submissions, papers out for review, those accepted/rejected for publication, etc. | 14 | Johnson (2019), Columbia University (2019), Editage (2018), Cochran (2016), Eve (2016), Gould (2009), Belcher (Unknown), WavesThink (2022), Stranack (2008), Shapiro (2005), PKP (2014), Smart (2018), Scholastica (2021), Jawaid (2017) |
| | 5. Suggests to apply for an International Standard Serial Number (ISSN) | 13 | Johnson (2019), University of Kansas (2022), Kumar (2021), Cochran (2016), VAD (2018), Freeman Jr (2018), Scholastica (2018), Stranack (2008), PKP (2014), Smart (2018), Scholastica (2021), Scholastica (2019), Jawaid (2017) |
| | 6. Suggests to register a Digital Object Identifier (DOI) for each article in a new journal | 13 | Johnson (2019), Kumar (2021), Cochran (2016), VAD (2018), Eve (2016), Solomon (2019), Scholastica (2018), Stranack (2008), PKP (2014), Smart (2018), Scholastica (2021), Scholastica (2019), Jawaid (2017) |
| | 7. Suggests to outline how a new journal will solicit content from researchers | 12 | University of Kansas (2022), Freeman Jr (2018), Belcher (Unknown), Trikha (2016), TRU Libraries (2021), Solomon (2019), CSE (2015), Stranack (2008), Shapiro (2005), Smart (2018), Scholastica (2021) Crammer (1998) |
| | 8. Suggests to determine the types of articles the new journal will publish (reviews, editorials, conference proceedings, etc.) | 9 | Kumar (2021), Cochran (2016), TRU Libraries (2021), Bond (2019), Stranack (2008), Smart (2018), Scholastica (2021), Card (2020), Jawaid (2017) |
| | 9. Suggests to determine out how often the new journal will publish | 15 | Kumar (2021), Editage (2018), Cochran (2016), Saunders (2017), VAD (2018), Eve (2016), Gould (2009), Belcher (Unknown), TRU Libraries (2021), Bond (2019), Scholastica (2018), Stranack (2008), Smart (2018), Scholastica (2019), Jawaid (2017) |
| | 10. Suggests to consult other professionals before starting a journal (institutional library, other publishers/publisher groups, experts in the field, etc. | 8 | Babar (2014), Kumar (2021), Freeman Jr (2018), TRU Libraries (2021), Scholastica (2018), CSE (2015), Scholastica (2019), Card (2020). |
| | 11. Suggests to detail whether the journal will be sponsored by an existing organization | 7 | Freeman Jr (2018), Gould (2009), TRU Libraries (2021), Solomon (2019), Scholastica (2018), Smart (2018), Scholastica (2019) |

**Table 7.** (*Continued*)

| Theme | Derived Unique Recommendation | Number of Sources Suggesting Recommendation | Source Identifier |
|---|---|---|---|
| **Editorial Review Processes** | 12. Suggest to set up an editorial board | 30 | Johnson (2019), University of Kansas (2022), University of Toronto (Unknown), Columbia University (2019), Babar (2014), Kumar (2021), Editage (2018), Cochran (2016), Saunders (2017), VAD (2018), Eve (2016), Freeman Jr (2018), Gould (2009), Booth (2015), Belcher (Unknown), Trikha (2016), TRU Libraries (2021), Solomon (2019), Scholastica (2018), CSE (2015), Stranack (2008), Shapiro (2005), PKP (2014), Smart (2018), Scholastica (2021), Scholastica (2019), Card (2020), Jawaid (2017), Crammer (1998), Antony (2004) |
| | 13. Suggests to create an editorial meeting structure involving reports of individual accountability, discussion of matters requiring decisions, etc. | 10 | Saunders (2017), Freeman Jr (2018), Trikha (2016), Scholastica (2018), Stranack (2008), Smart (2018), Scholastica (2018), Card (2020), Crammer (1998), Antony (2004) |
| | 14. Suggests to create standards to evaluate articles for inclusion | 8 | University of Kansas (2022), Freeman Jr (2018), Smart (2018), Scholastica (2021), Scholastica (2019), Card (2020), Crammer (1998), Antony (2004) |
| | 15. Suggests to detail how long editor terms will be, and what the process will be for replacement/reappointment of editorial board members | 5 | Freeman Jr (2018), TRU Libraries (2021), Stranack (2008), Smart (2018), Smart (2018), Jawaid (2017) |
| | 16. Suggests to detail whether training of editorial staff will be done | 5 | Freeman Jr (2018), Solomon (2019), Shapiro (2005), Smart (2018), Card (2020) |
| | 17. Suggests to detail how journal will deal with corrections and erratum | 1 | Smart (2018) |
| **Peer Review Processes** | 18. Suggests to create a detailed protocol for peer review detailing how many peer-reviewers per article, the specific criteria, the process of solving discrepancies among reviewers, etc. | 13 | Kumar (2021), Gould (2009), Belcher (Unknown), TRU Libraries (2021), Solomon (2019), Scholastica (2018), Stranack (2008), Smart (2018), Scholastica (2021), Scholastica (2019), Card (2020), Jawaid (2017), Antony (2004) |
| | 19. Suggests to reach out to potential reviewers who are working closely in the field of the article | 12 | Johnson (2019), Kumar (2021), Eve (2016), Freeman Jr (2018), Gould (2009), Belcher (Unknown), Trikha (2016), Stranack (2008), Smart (2018), Scholastica (2021), Card (2020), Jawaid (2017) |
| | 20. Suggests to set a timeline for peer reviewers and when they should send in their feedback for the papers by | 9 | Babar (2014), Kumar (2021), Gould (2009), Belcher (Unknown), TRU Libraries (2021), Stranack (2008), Smart (2018), Scholastica (2021), Scholastica (2019) |
| | 21. Suggests to send personal, customized requests to potential reviewers | 6 | Johnson (2019), Kumar (2021), Eve (2016), Freeman Jr (2018), Gould (2009), Belcher (Unknown) |
| | 22. Suggests to make distinctions indicative of your journal not being predatory (not asking for money, etc.) | 4 | Johnson (2019), Columbia University (2019), Freeman Jr (2018), CSE (2015) |
| | 23. Suggests to give reviewers monetary compensation/honorarium as an incentive for review | 1 | Gould (2009) |
| | 24. Suggests specific plagiarism-checking software to ensure originality of submissions to new journal | 1 | Jawaid (2017) |
| **Open Access Publishing** | 25. Suggests the use of specific software to set up and manage open access journals (Open Journal Systems, etc.) | 13 | Johnson (2019), University of Toronto (Unknown), Kumar (2021), Eve (2016), Freeman Jr (2018), Gould (2009), Bond (2019), WavesThink (2022), Stranack (2008), Shapiro (2005), PKP (2014), Smart (2018), Swan (2012) |
| | 26. Suggests to decide whether new journal will be open-access, available only by subscription, or a combination | 13 | University of Kansas (2022), Columbia University (2019), Saunders (2017), Eve (2016), Freeman Jr (2018), Gould (2009), CSE (2015), Stranack (2008), Shapiro (2005), Smart (2018), Swan (2012), Scholastica (2021), Scholastica (2019) |

(*Continued*)

**Table 7.** (Continued)

| Theme | Derived Unique Recommendation | Number of Sources Suggesting Recommendation | Source Identifier |
|---|---|---|---|
| **Copyediting/ Typesetting** | 27. Suggests to confirm whether copyediting and typesetting will be outsourced or be done in-house | 12 | Kumar (2021), Eve (2016), Freeman Jr (2018), Belcher (Unknown), Trikha (2016), Solomon (2019), Scholastica (2018), Stranack (2008), Shapiro (2005), Smart (2018), Jawaid (2017), Antony (2004) |
| | 28. Suggests the use of specific software to copyedit and typeset articles (QuarkXpress, Adobe FrameMaker, Adobe InDesign, etc.) | 5 | Johnson (2019), Belcher (Unknown), Stranack (2008), Jawaid (2017), Antony (2004) |
| | 29. Suggests to confirm whether the use of a journal management software charges for copyediting, proofreading, and typesetting | 1 | Columbia University (2019) |
| **Production** | 30. Suggests to decide on the format of your journal (print, online, etc.) | 18 | University of Kansas (2022), Kumar (2021), Cochran (2016), Saunders (2017), Freeman Jr (2018), Gould (2009), Belcher (Unknown), TRU Libraries (2021), Bond (2019), Scholastica (2018), Stranack (2008), Shapiro (2005), Smart (2018), Scholastica (2021), Scholastica (2019), Card (2020), Jawaid (2017), Antony (2004) |
| | 31. Suggests to decide where the journal will be published and/or printed | 13 | University of Kansas (2022), Columbia University (2019), Saunders (2017), Freeman Jr (2018), Belcher (Unknown), Scholastica (2018), Stranack (2008), Shapiro (2005), Smart (2018), Scholastica (2021), Card (2020), Jawaid (2017), Antony (2004) |
| | 32. Suggests to create a Memorandum of Agreement/ Understanding that spells out expectations for the journal and the printer | 2 | University of Kansas (2022), Freeman Jr (2018) |
| | 33. Suggests to confirm the publishing format (HTML, PDF, JATS XML, etc.) | 11 | Kumar (2021), Saunders (2017), VAD (2018), Gould (2009), TRU Libraries (2021), Scholastica (2018), PKP (2014), Smart (2018), Scholastica (2021), Scholastica (2019), Antony (2004) |
| **Indexing/ Archiving/ Metrics** | 34. Suggests specific platforms to index journals (SCOPUS, PubMed, Web of Science, ISI, DOAJ, etc.) | 11 | Johnson (2019), Editage (2018), Cochran (2016), Saunders (2017), VAD (2018), Scholastica (2018), WavesThink (2022), Stranack (2008), Smart (2018), Jawaid (2017), Antony (2004) |
| | 35. Suggests to deposit a DOI for an article on indexing services (Crossref, etc.) | 7 | Cochran (2016), VAD (2018), Scholastica (2018), Stranack (2008), Smart (2018), Scholastica (2021), Jawaid (2017) |
| | 36. Suggests specific avenues for journal file archival (university library network, off-campus private storage providers, etc.) | 3 | Gould (2009), Stranack (2008), Smart (2018) |
| | 37. Suggests specific journal metric platforms to enrol your journal in (Google Analytics, Altmetric Reporting, Clarivate Analytics, etc.) | 3 | Scholastica (2018), Stranack (2008), Jawaid (2017) |
| | 38. Suggests ways to bookmark journal article webpages and combat "link rot" | 1 | Gould (2009) |
| | 39. Suggests to index on a platform that promises to provide an impact factor | 1 | Smart (2018) |

*(Continued)*

**Table 7.** (Continued)

| Theme | Derived Unique Recommendation | Number of Sources Suggesting Recommendation | Source Identifier |
|---|---|---|---|
| **Marketing/ Promotion** | 40. Suggests to outline how your journal will advertise | 10 | University of Kansas (2022), Cochran (2016), Freeman Jr (2018), Belcher (Unknown), Solomon (2019), Stranack (2008), Shapiro (2005), PKP (2014), Smart (2018), Scholastica (2021) |
| | 41. Suggests specific ways your journal can advertise (call for papers flyer, email campaign, conference promotions, etc.) | 8 | Cochran (2016), VAD (2018), Freeman Jr (2018), Belcher (Unknown), CSE (2015), Stranack (2008), Smart (2018), Scholastica (2021) |
| | 42. Suggests the use of social media to improve publicity | 6 | Johnson (2019), Babar (2014), Cochran (2016), VAD (2018), Smart (2018), Scholastica (2021) |
| | 43. Suggests the use of networks at institutional departments to improve publicity | 4 | Johnson (2019), Babar (2014), Stranack (2008), Smart (2018) |
| | 44. Suggests to keep an independent distribution/ subscriber mailing list | 3 | Saunders (2017), Belcher (Unknown), Stranack (2008) |
| | 45. Suggests to conduct a survey of prospective readership to inform journal decisions | 1 | Saunders (2017) |
| **Funding Models** | 46. Suggests to indicate how the journal will be funded and sustained (financial model) | 23 | University of Kansas (2022), University of Toronto (Unknown), Kumar (2021), Saunders (2017), Eve (2016), Freeman Jr (2018), Gould (2009), Booth (2015), Belcher (Unknown), Trikha (2016), TRU Libraries (2021), Solomon (2019), Bond (2019), Scholastica (2018), CSE (2015), Stranack (2008), Shapiro (2005), PKP (2014), Smart (2018), Swan (2012), Scholastica (2021), Scholastica (2019), Jawaid (2017) |
| | 47. Suggests to try and secure institutional support | 10 | Saunders (2017), Freeman Jr (2018), Gould (2009), Scholastica (2018), CSE (2015), Stranack (2008), Smart (2018), Scholastica (2021), Scholastica (2019), Jawaid (2017) |
| | 48. Suggests that article processing charges can help raise funds to grow your journal | 8 | Kumar (2021), Bond (2019), CSE (2015), Stranack (2008), Smart (2018), Swan (2012), Scholastica (2021), Scholastica (2019) |
| | 49. Suggests to find out whether researchers can bypass article processing charges by using grants or submitting a waiver | 1 | Columbia University (2019) |
| | 50. Suggests to have an adequate independent bookkeeping and auditing system in place to track subscription costs | 1 | Saunders (2017) |
| | 51. Suggests to gather funding to last at least 5–6 issues before starting the journal | 1 | Trikha (2016) |

Abbreviations: TRU Libraries = Thompson Rivers University Libraries, VAD = Vereinigung für Afrikawissenschaften in Deutschland, PKP = Public Knowledge Project, CSE = Council of Science Editors

determine where the journal will be published and/or printed (n = 13 sources), and to confirm the publishing format (e.g., HTML, PDF, JATS, XML) (n = 11 sources).

**Indexing/Archiving/Metrics.** In the indexing/archiving/metrics theme, eleven sources recommended to utilize specific platforms such as Scopus, PubMed, Web of Science, and Institute for Scientific Information (ISI) to index journals, while seven sources recommended to deposit a DOI for an article on indexing services such as Crossref. Notably, only one source recommended to index the journal on a platform that provides an impact factor.

**Marketing/Promotion.** The most common recommendation in the marketing/promotion theme was to outline how the new journal will be advertised (n = 10 sources). Eight sources recommended to utilize specific methods to advertise the new journal (e.g., call for papers flyer, email campaign, conference promotions). More specifically, six sources indicated

to utilize social media to improve publicity, while four sources recommended to utilize connections at institutional departments to improve publicity.

**Funding and funding models.** One of the most common recommendations across all themes was to indicate how the new journal will be funded and sustained (n = 23 sources). Other common recommendations were to secure institutional support (n = 10 sources), and to raise funds for the new journal through article processing charges (n = 8 sources). Of the included sources, only seven considered the potential barriers and/or facilitators that publishers may face when implementing their recommendations. The most common barriers mentioned were limited financial resources and the significant time commitment required for creating and operating a scholarly journal.

## Discussion

### Significance of findings

The aim of this scoping review was to identify and describe existing recommendations for the creation of a new scholarly journal. This review identified 33 sources providing guidance on scholarly journal creation, with corresponding authors and organizations from 9 countries. A total of 51 unique recommendations were extracted from the 33 sources. Upon examining these recommendations, two clear patterns emerged. First, only 10 of the included sources positing recommendations were informed by evidence-based research or rigorously designed studies. In contrast, 23 sources provided recommendations based on expert opinion or personal experience in the scholarly publishing field. Secondly, while a large variety of recommendations were provided, many lacked details. For example, while nearly half of all sources indicated to build a website for a new journal, only five sources recommended actionable steps to build a website (e.g., purchase a domain name, find a web-hosting platform). Out of 33 sources, 30 recommended to set up an editorial board. However, only five sources recommended to devise a process for replacement/reappointment of editorial board members, or detail whether editorial staff will be trained. This lack of detail in recommendations was observed across all nine themes.

### Comparative literature

To our knowledge, no scoping review has identified and mapped existing evidence-based guidance for starting a new scholarly journal. However, relevant past studies can serve as comparative literature. A 2021 literature review detailed notable policies, standards, and logistical considerations for scholarly publishers [5]. The article detailed information related to journal identifiers (e.g., ISSN and DOI), recommended editorial board policies, peer review and reviewer guidelines, journal hosting platforms, and access policies. The author noted that a thorough understanding of the current landscape of scholarly publishing can aid editors in identifying key policies and guidelines to abide by. The author also noted that standards vary internationally, and there is no single, consolidated source that publishers can refer to for journal creation and maintenance [5]. In a systematic review and meta-analysis published in December 2021, researchers investigated the role of clinicians from the perspective of publishers [25]. Based on the results, researchers devised publication recommendations with regards to the responsibilities of authors, editors, and reviewers. Primary recommendations included to set up an editorial board with detailed structure for evaluation, and to create a comprehensive protocol for peer review [25]. In a commentary published in the International Journal of Clinical Practice, members of the Blackwell Publishing group summarized major recommendations and principles of academic publishing to provide practical guidance that editors can adopt [26]. Major recommendations included ensuring a detailed peer-review process that

minimizes bias, maintaining a high-standard of peer reviewer selection, and maintaining transparency and research integrity [26]. A similar 2018 review article published in the Journal of Research Initiatives detailed an editor's perspective to starting an open-access journal [9]. Major recommendations included finding the journal's unique niche, determining the administrative and editorial structure of the journal, evaluating funding for the journal, and marketing journal content. It is of note that this article detailed their primary method of review as Scholarly Personal Narrative (SPN), with the majority of recommendations based in non-research-based literature with first-hand commentary lacking evidence [9]. All of these findings similarly reflect what was observed in this study.

## Implications

Not all publishers are created equally; differences in funding, resources and country of origin can arguably all serve to advantage or disadvantage a publisher's ability to produce high-quality publication even if their editorial team has the best intentions. For example, one source in our review recommended offering peer reviewers monetary compensation (e.g., honorarium) as an incentive to reduce reviewer delays [27]. Another source suggested offering incentives to reviewers, such as publishing their name in an annual list of reviewers in the journal, offering free copies of the journal or access to its website, or providing gift vouchers [28]. The same source also acknowledged that while offering reviewers payment is uncommon due to how expensive the service is, it may be necessary in some circumstances [28]. These approaches are not likely equitable in terms of feasibility across the diverse range of publishers [29]. More recently, one approach some journals have taken to incentivize reviewers is to offer tokens/credits which may be redeemed by reviewers as discounts towards publishing their next article in the same journal [30]. However, this is dependent on the size of the journal's field, as smaller fields would likely make this prohibitive. In contrast, other sources strongly disagreed, recommending that editors, copyeditors, reviewers, and members of the advisory board all be recruited on a voluntary basis to minimize expenses for the new journal [9, 31, 32]. As well, we found that the range of guidance offered greatly differed, suggesting a lack of adequate information about all aspects of the scholarly publishing process. For example, the journal operations theme received a total of 137 recommendations while the funding models theme only received 45 recommendations. In addition, important considerations when starting a new journal such as applying/registering for membership in publishing organizations (e.g., DOAJ (in the case of open access journals), OASPA, COPE) as well as information about copyright information was rarely observed. However, as this is done on the publisher level rather than the journal level, any organization that has an existing journal would not need to go through this process [29].

Scholarly journals serve to enable communication and exchange between researchers interested in the same topic [33]. Certain readers, such as patients, early career researchers and graduate students, trust and expect these journals to have enforced a rigorous peer-review of the findings presented to them [34]. Similarly, authors expect to receive valuable feedback from experts regarding their work when submitting to these journals [34]. New journals facilitating these conversations must also engage in transparent and ethical publishing practices in order to uphold the reporting of robust and valid research. Given the directly contrasting recommendation and the difference in the range of advice offered to new journals that we found in this review, a consensus approach on how to launch and operate a new scholarly journal may better support the quality of scholarly publishing in new journals.

Even upon successful launching a new journal, there are more obstacles to face when it comes to sustaining the periodical, such as establishing credibility, overcoming a lack of

visibility, and developing a sustainable financial model [32, 35]. Credibility is of great importance to scholarly publishing; it is what attracts authors to submit their manuscripts to specific journals and in turn, what draws readership. Credibility is also built with time, which new journals do not have, and a problem established journals face less often. For a new journal, establishing credibility may be one of its biggest challenges [9]. One way a new journal can achieve this is through the recruitment of an effective and respected editorial board, which not only lends credibility from the journal's onset, but can also help attract submissions and reviewers [27, 31]. It should be noted that although having a reputable editorial board may be an excellent starting point in building credibility, implementing this recommendation it is not sufficient to warrant the credibility of a scholarly journal. Similarly, implementing any of the recommendations identified in this review in isolation does not guarantee the successful operations of a new scholarly journal. Instead, the careful consideration of the entire body of recommendations should be considered by publishers to ensure considering all specific areas instrumental to success when creating and running a high-quality biomedical scholarly journal. Our review identified 59 recommendations regarding the editorial review processes theme, making it the second-most discussed theme across our included sources. This suggests an emphasis is placed on the role of an editorial board in existing guidance for new scholarly journals.

The next obstacle a new journal must overcome is its lack of visibility, availability, and readership, which may be increased with indexing services [36]. Due to the vetting process reputable bibliographic databases employ, indexed journals are generally considered to be of higher quality than non-indexed journals [36, 37]. Once indexed, the journal's contents typically become more widely accessible, increasing its readership while also improving the journal's reputation as a reliable source [36]. This review identified 26 recommendations (across 11 sources) in the Indexing/Archiving/Metrics theme, with most recommendations suggesting indexing platforms to which publishers of new journals should apply. Of these 11 sources, only 2 sources provided details beyond what platforms journals may be indexed in. These details included qualifying criteria (i.e., a regular publishing schedule, a respected editorial board, a peer review process) [35], and steps new journals must take before applying to indexing services (e.g., applying for an ISSN, depositing DOIs with Crossref) [38].

The last major obstacle publishers may face is in developing a sustainable source of funding to ensure their new journal's long-term viability [35]. Considering a new journal needs time to establish credibility and qualify for indexing databases, a reliable source of funding is necessary to support the journal during these processes. A total of 45 recommendations regarding the funding models' theme was identified by this review. Most recommendations in this theme either suggested to secure institutional funding or to charge article processing fees to raise funds for the new journal. However, few details were provided beyond those initial recommendations.

Aside from these obstacles, small publishers must also compete with very large and well-established publishing houses in multiple areas. Despite the lower cost of publishing electronically compared to traditional print mediums, very large publishers still hold the advantage financially. These publishers can more easily offer their journals at a discount through bundling, while a small publisher or a scholarly association is less likely to have as many journals and thus, cannot offer the same bundle sizes [33]. It would likely be more difficult for a small publisher or scholarly association to offset profit losses like these than for a larger publisher. Furthermore, small publishers must also overcome their new journal's lack of discoverability. For a small publisher, generating interest in their new journal is difficult without an already existing archive of content [39]. In contrast, large publishing houses already dominate the scholarly publishing field, each comprising numerous journals. To illustrate, Elsevier currently

publishes over 2800 active journals [40], while SAGE Publishing publishes more than 900 journals at present [41]. Unlike larger publishers, a small publisher may not have the resources necessary to expand their journal's presence. With these resources, a publisher may invest in publishing software, develop overlay tools, and fully utilize marketing opportunities presented by social media platforms [39]. Such barriers will only be overcome through the journal's growth, which for new journals takes an extensive amount of time and funds to accomplish.

**Future directions.** At present, this scoping review has identified 51 unique recommendations pertaining to the development of a new scholarly journal. Notably, only 7 of the 33 sources included in this review mentioned potential barriers and/or facilitators that publishers may face when implementing their recommendations. Of these 7 sources, none provided detailed explanations of how a publisher may overcome these barriers. To mitigate these cumulative barriers, we propose that the recommendations identified by this review be further explored to inform the development of a guideline regarding starting a new scholarly biomedical journal. Once a list of all extracted recommendations is compiled, relevant stakeholders (e.g., publishers, academic researchers with expertise in publication science) would be sought and invited to participate in a Delphi survey to agree upon guidance. Moreover, while outside of the scope of the present review, future research could include contacting different publishers and requesting that they share for analysis their internal documents for the creation of a new scholarly journal for comparative analysis.

Such a guideline may serve to better standardize the best practices, and criteria required to launch a successful, credible journal. It may also help streamline the resource implications and process of launching a journal for large and small publishers alike. Establishing evidence-based criteria towards journal creation not only allows new publishers to avoid engaging in dubious scholarly publishing practices but may also help reduce authors' submissions to predatory journals by providing them with a basis to differentiate legitimate journals from 'predatory' ones. One such tool currently in development is the Journal Transparency tool, which seeks to provide users with information about a journal's operations and transparency practices [42, 43]. Furthermore, a related future research direction worth exploring may be to study the operational modalities, tools, and methodologies these predatory journals employ to recruit new, inexperienced researchers, and in the publicization of their journal. The resulting identified tools and strategies may prove useful in informing the development of a guideline for journal creation. The establishment of a guideline may also enable more ideas and discoveries to be heard from local researchers, while also helping to close the knowledge gap between the voices of authors in developed and developing nations. Although such a guideline would be beneficial to both big and small publishers, we do recognize that small publishers are likely to have more difficulty in implementing and following such a guideline, as compared to big publishers. For this guideline to be applicable and feasible, it must provide information on any anticipated barriers and facilitators to its implementation, strategies to mitigate these barriers, and clearly state any resource implications of applying the guideline.

## Strengths and limitations

We screened titles/abstracts, as well as conducted data extraction, independently and in duplicate, which serves as a major strength of this study. We also searched multiple bibliographic databases and grey literature sources, and had our search strategies peer reviewed by an additional academic librarian which is another strength of our methodological approach.

With respect to limitations, it is possible that we missed eligible sources that mentioned recommendations for creating a biomedical journal but do not contain key words relating to our research question in their title or abstract. Given the nature of our research question and

following consultation with an academic librarian, it was determined that the best search strategy would have been a broad one. This, therefore, came with the drawback of us needing to screen more records. Screening titles and abstracts was the most feasible way to initially process the list of potentially viable articles. Although not including libraries may reflect a limitation of our study, we believe that our search strategy best balanced project comprehensiveness and practicality. The study's limitations also include the fact that only sources written in or spoken in English were considered. We found that to overcome the dominance of large international journals, many non-English speaking countries such as Brazil, China, and India have increased the number of publicly funded national journals available for their scientists to publish in [4]. Due to the growth and emergence of numerous non-English journals in the past two decades, we may have missed articles that contain recommendations on starting a biomedical journal from these sources [4]. Furthermore, while we sought both publicly available search results, as well as those available to us via our university library system and interlibrary loan, we acknowledge that this review will not have captured guidance found in internal/unpublished documents, created by consultancy companies, or from proprietary/confidential material created by publishers. As well, considering how limited the structured research on this topic is, very few evidence-based recommendations would have been formulated and subsequently, extracted through this study.

## Conclusion

The purpose of this scoping review was to identify and describe existing guidelines on starting a new scholarly journal. The majority of existing recommendations placed an emphasis on technical journal operations and the role of an editorial board in journal creation. Most of the posited recommendations lacked evidence or were not informed by rigorously designed studies, indicating that there are no standardized criteria for new journal creation and their publishing practices. The range of guidance also differed greatly, suggesting a lack of conclusive information about the scholarly publishing process. Given the contrasting recommendations and varying range of advice, a stakeholder-led, survey-informed consensus approach on creating and operating a new scholarly journal may better support the quality of scholarly publishing in new journals. The establishment of such evidence-based guidelines for journal creation may allow for the standardization of high-quality publication practices across all publishers.

## Supporting information

**S1 Checklist. Preferred Reporting Items for Systematic reviews and Meta-Analyses extension for Scoping Reviews (PRISMA-ScR) checklist.**
(DOCX)

## Acknowledgments

Peer review using PRESS was completed by Nigèle Langlois, Research Librarian, Health Sciences Library, University of Ottawa.

## Author Contributions

**Conceptualization:** Jeremy Y. Ng, Kelly D. Cobey.

**Data curation:** Jeremy Y. Ng, Kelly D. Cobey.

**Formal analysis:** Jeremy Y. Ng, Kelly D. Cobey, Saad Ahmed, Valerie Chow, Sharleen G. Maduranayagam, Lucas J. Santoro.

**Funding acquisition:** Jeremy Y. Ng.

**Investigation:** Jeremy Y. Ng, Kelly D. Cobey.

**Methodology:** Jeremy Y. Ng, Kelly D. Cobey, Lindsey Sikora, Ana Marusic, Daniel Shanahan, Randy Townsend, Alan Ehrlich, Alfonso Iorio, David Moher.

**Supervision:** Jeremy Y. Ng, Kelly D. Cobey, David Moher.

**Writing – original draft:** Jeremy Y. Ng.

**Writing – review & editing:** Kelly D. Cobey, Saad Ahmed, Valerie Chow, Sharleen G. Maduranayagam, Lucas J. Santoro, Lindsey Sikora, Ana Marusic, Daniel Shanahan, Randy Townsend, Alan Ehrlich, Alfonso Iorio, David Moher.

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
