## [Decision Letter · Decision Letter 0]

7 Dec 2022

PONE-D-22-21604Recommendations and Guidelines for Creating Scholarly Journals: A Scoping ReviewPLOS ONE

Dear Dr. Jeremy Y. Ng

Thank you for submitting your manuscript to PLOS ONE. After careful consideration, we feel that it has merit but does not fully meet PLOS ONE’s publication criteria as it currently stands. Therefore, we invite you to submit a revised version of the manuscript that addresses the points raised during the review process.

Please submit your revised manuscript by Jan 21 2023 11:59PM If you will need more time than this to complete your revisions, please reply to this message or contact the journal office at plosone@plos.org. Please include the following items when submitting your revised manuscript:

We look forward to receiving your revised manuscript.

Kind regards,

Shahabedin Rahmatizadeh, Ph.D.

Academic Editor

PLOS ONE

Journal Requirements:

Reviewers' comments:

Reviewer's Responses to Questions

**Comments to the Author**

1. Is the manuscript technically sound, and do the data support the conclusions?

Reviewer #1: Yes

Reviewer #2: Yes

Reviewer #3: Yes

2. Has the statistical analysis been performed appropriately and rigorously? 

Reviewer #1: Yes

Reviewer #2: Yes

Reviewer #3: Yes

3. Have the authors made all data underlying the findings in their manuscript fully available?

Reviewer #1: Yes

Reviewer #2: Yes

Reviewer #3: Yes

4. Is the manuscript presented in an intelligible fashion and written in standard English?

Reviewer #1: Yes

Reviewer #2: Yes

Reviewer #3: Yes

5. Review Comments to the Author

Reviewer #1: The authors have done a credible job in writing this manuscript. I have the following suggestions to encourage authors to inspire thought in addressing the following:

1. Authors may build this in the background:

-What is the rationale for having more journals despite having existing ones?

-Given that there are practices developed by COPE, DOAJ and others, why is there a further need for having these guidelines?

2. It might be helpful for the readers to have the findings organized in a tabular format with having identified the unique features as well as loopholes in the existing guidelines.

3. It would have also been helpful to add a brief section on ethically publishing journals, given that there is a hoard of scavenger journals, old and new, identified every year.

4. Authors may take the liberty of adding a recommendations section in response to the existing loopholes, especially given that it is a scoping review and may lend some room to the same.

-For example, having a reputed editorial board and indexing may be effective but several times it does not necessarily warrant credibility of a scholarly journal.

5. Would having a global guideline for all publishers, big and small, do justice? This may be a scope for this manuscript given that authors wish to work on recommendations.

Reviewer #2: The purpose of this paper is to answer the research question “What recommendations exist for starting a scholarly biomedical journal”. It approaches this research question by engaging in a literature review of both primary sources and gray literature in the English language. The review first isolated relevant sources, filtered according to language (English) and relevance. The results of the review are broken down from there into discrete recommendations which were then de-duplicated and coded by theme.

I found the overall methodology compelling and the granularity with which the process was laid out to be adequate and appropriate. The authors were meticulous in their study design, including in their registering the study with the OSF. The writing in the paper was clear and I found nearly all of my questioned answered as the paper progressed. In particular, the inclusion of the full search strategy, including queries by database, provided important data for the reader and for repeatability.

There are only a few minor areas in which I think the authors should consider adding changes. The first involves the difference between the stated research question, noted above, and this statement in the abstract: “This scoping review aimed to identify and describe existing guidelines for starting a biomedical scholarly journal.”. The differences in these statements, including the difference between guidelines and recommendations. As stated on page 4 “Guidelines and Recommendations: A CDC Primer” (https://stacks.cdc.gov/view/cdc/81408/cdc_81408_DS1.pdf, cited in the document as coming from the WHO handbook for guideline development, although the link in the report is inactive), “Guidelines are documents that contain recommendations about health interventions whether they be clinical, public health, or policy recommendations. Recommendations provide information about what policy makers, health care providers, or patients should do. Recommendations imply choices between different interventions that have an impact on health and that have ramifications for the use of resources.” The difference is significant enough that I believe the authors should just use the research question in the abstract.

Given that the researchers opted to restrict their search methodology to title and abstract searches, I think they should spend more time describing why they made that choice rather than utilizing full text searching to locate articles that may contain recommendations that are relevant even if journal publishing is not the main thrust of the article in question. I suspect I know why this choice was made, but I would like verification from the authors rather than relying on my own assumptions.

Because this paper limited the literature review to English journals, by necessity it missed literature that may exist on this subject from regions such as Africa and the Indian subcontinent where we see significant numbers of new journals emerging. While all studies need to be scoped for a variety of reasons, I would have liked to have seen the authors make more of a nod to that limitation, rather than just noting it as a limitation of the language.

Another thing worth noting is that some of the descriptions of the operational modalities of predatory journals may actually be instructive in constructing guidelines in this case. While such journals are not a healthy mechanism for scientific discovery, they have been successful in recruiting unsuspecting new researchers and in publicizing themselves. A future direction for this research may involve studying the tools and methods those journals employ to see if those methodologies could be turned to more constructive purposes. Such a direction may be worth a nod in the conclusion.

I do not see sources 7, 16, 17, 18, 19, 20, 21, or 35 noted in the body of the text.

In addition to the data included in the appendices the authors should consider uploading RIS (or a similar format) files containing raw and distilled literature search results to OSF.

With those minor points made I feel like this is a high-quality paper that covers an important subject using a sound methodology. I recommend it for publication with minor revisions.

Reviewer #3: The idea of this paper is both somewhat interesting and relevant, albeit probably only to a limited group of professional: To identify and describe any existing guidelines for starting a scholarly journal. And to investigate whether this can be done through a scoping review methodology.

1) The title promises a scope related to creating scholarly journals as such but only delivers on biomedical journals. Perhaps it could be emphasized in the subtitle that this is indeed the point of the scoping review?

2) Sources: It was curious to see learn that the authors have chosen a host of classical medical science databases for this study, since this would naturally limit identified sources to those fields, while other sources external to medicine might also contain relevant information. The authors could add a little more as to the reasoning behind the choice of primary databases and what this means to potential biases. Also, the authors could spend a few lines arguing why they have not included libraries as a source of relevant information, or potentially information from publishers internal sources.

3) Method: Giving the comments above, it is my impression that the authors put too much emphasis on their methodology when discussing their points and in reaching conclusions. Statistically they are dealing with very low numbers and sources of limited riceness. While the scoping review has been carried out well it seems, its use in discussing the topic is clearly limited.

4) Results: It is hardly surprising that very few sources are identified (33), but perhaps striking that more than 5000 unique sources are obtained initially. Given the subject field and databases etc. It is to be expected that relevant titles are not abundant.

5) Conclusions: I would recommend that the authors consider the possible effects on their conclusions of not having access to publishers' internal information on starting journals. This would probably require a more qualitative approach, however.

6. PLOS authors have the option to publish the peer review history of their article (what does this mean?). If published, this will include your full peer review and any attached files.

Reviewer #1: **Yes: **Pragya Lodha

Reviewer #2: **Yes: **Jason Bengtson

Reviewer #3: No

---

## [Author Response · Author response to Decision Letter 0]

12 Jan 2023

Response to Reviewers

Dear Editor:

Thank you for your email. I am pleased to hear that you believe that our manuscript will have the potential to be published in PLOS ONE following requested revisions.

As per your request, please find our response to the peer-reviewers’ comments with every change outlined point by point below:

• We kindly thank the editor for providing their feedback on our manuscript.

Journal Requirements:

• We have ensured that our manuscript meets all PLOS ONE style requirements.

• We will make all data available at the DOI currently listed under the “Availability of Data and Materials” heading of our manuscript.

• No supplementary files are provided with our manuscript submission, and all of our raw data files are available on OSF as described in our original submission. We also converted our appendices into tables, and renumbered our tables accordingly.

• We have reviewed the reference list to ensure it is complete and correct. 

• We have added following references: 

o Rallison SP. What are Journals for? Annals of the Royal College of Surgeons of England. 2015 Mar;97(2):89-91. doi: 10.1308/003588414X14055925061397 

o Meneghini R. Emerging journals. EMBO reports. 2012;13(2):106-108. doi: 10.1038/embor.2011.252. 

o Grudniewicz A, Moher D, Cobey KD, Bryson GL, Cukier S, Allen K, et al. Predatory journals: No definition, no defence. Nature. 2019 Dec;576(7786):210-212. https://doi.org/10.1038/d41586-019-03759-y.

• We have removed the following references:

o Carpenter CR, Cone DC, Sarli CC. Using Publication Metrics to Highlight Academic Productivity and Research Impact. Acad Emerg Med. 2014 Oct;21(10):1160–72. https://doi.org/10.1111/acem.12482

Reviewers' comments:

Reviewer's Responses to Questions

Comments to the Author

1. Is the manuscript technically sound, and do the data support the conclusions?

Reviewer #1: Yes

Reviewer #2: Yes

Reviewer #3: Yes

2. Has the statistical analysis been performed appropriately and rigorously? 

Reviewer #1: Yes

Reviewer #2: Yes

Reviewer #3: Yes

3. Have the authors made all data underlying the findings in their manuscript fully available?

Reviewer #1: Yes

Reviewer #2: Yes

Reviewer #3: Yes

4. Is the manuscript presented in an intelligible fashion and written in standard English?

Reviewer #1: Yes

Reviewer #2: Yes

Reviewer #3: Yes

5. Review Comments to the Author

Reviewer #1: The authors have done a credible job in writing this manuscript. I have the following suggestions to encourage authors to inspire thought in addressing the following:

• We kindly thank this reviewer for providing their feedback on our manuscript.

1. Authors may build this in the background:

-What is the rationale for having more journals despite having existing ones?

• We agree with this suggestion and have briefly discussed the rationale for the creation of new journals within the Background of the manuscript.

-Given that there are practices developed by COPE, DOAJ and others, why is there a further need for having these guidelines?

• Guidance in this field was issued already by COPE, DOAJ, OASPA, and WAME, but there is not (yet) a broad consensus in this space. The purpose of this review is to identify and categorize all recommendations that exist both within and outside of the above organizations, something that prior to this review was unknown

2. It might be helpful for the readers to have the findings organized in a tabular format with having identified the unique features as well as loopholes in the existing guidelines.

• We agree and have revised Table 7 to now include the 51 unique recommendations and the number of sources suggesting each unique recommendation, to display our findings more clearly in the manuscript. We have also updated all relevant details in the table legend section to reflect this change.

3. It would have also been helpful to add a brief section on ethically publishing journals, given that there is a hoard of scavenger journals, old and new, identified every year.

• We have included a brief section on ethically publishing and predatory journals (we believe this is what the reviewer means by “scavenger journals”), and the importance of understanding predatory publishing during the creation of a new scholarly journal.

4. Authors may take the liberty of adding a recommendations section in response to the existing loopholes, especially given that it is a scoping review and may lend some room to the same.

-For example, having a reputed editorial board and indexing may be effective but several times it does not necessarily warrant credibility of a scholarly journal.

• We have included a brief statement regarding existing loopholes in the Discussion section.

5. Would having a global guideline for all publishers, big and small, do justice? This may be a scope for this manuscript given that authors wish to work on recommendations.

• We recognize that small publishers are likely to have more difficulty following such a global guideline compared to big publishers. We have now acknowledged this in the Future Directions section.

Reviewer #2: The purpose of this paper is to answer the research question “What recommendations exist for starting a scholarly biomedical journal”. It approaches this research question by engaging in a literature review of both primary sources and gray literature in the English language. The review first isolated relevant sources, filtered according to language (English) and relevance. The results of the review are broken down from there into discrete recommendations which were then de-duplicated and coded by theme.

• We kindly thank this reviewer for providing their feedback on our manuscript.

I found the overall methodology compelling and the granularity with which the process was laid out to be adequate and appropriate. The authors were meticulous in their study design, including in their registering the study with the OSF. The writing in the paper was clear and I found nearly all of my questioned answered as the paper progressed. In particular, the inclusion of the full search strategy, including queries by database, provided important data for the reader and for repeatability.

There are only a few minor areas in which I think the authors should consider adding changes. The first involves the difference between the stated research question, noted above, and this statement in the abstract: “This scoping review aimed to identify and describe existing guidelines for starting a biomedical scholarly journal.”. The differences in these statements, including the difference between guidelines and recommendations. As stated on page 4 “Guidelines and Recommendations: A CDC Primer” (https://stacks.cdc.gov/view/cdc/81408/cdc_81408_DS1.pdf, cited in the document as coming from the WHO handbook for guideline development, although the link in the report is inactive), “Guidelines are documents that contain recommendations about health interventions whether they be clinical, public health, or policy recommendations. Recommendations provide information about what policy makers, health care providers, or patients should do. Recommendations imply choices between different interventions that have an impact on health and that have ramifications for the use of resources.” The difference is significant enough that I believe the authors should just use the research question in the abstract.

• We agree and have removed instances of the word “guidelines” in our manuscript.

Given that the researchers opted to restrict their search methodology to title and abstract searches, I think they should spend more time describing why they made that choice rather than utilizing full text searching to locate articles that may contain recommendations that are relevant even if journal publishing is not the main thrust of the article in question. I suspect I know why this choice was made, but I would like verification from the authors rather than relying on my own assumptions.

• Given the nature of our research question and following consultation with an academic librarian, it was determined that the best search strategy would be a broad one. This, therefore, required us needing to screen more records. Screening titles and abstracts was the most feasible way to initially process the list of potentially viable articles. We should also highlight that our search strategy was also peer reviewed (PRESSed) by another academic librarian to enhance its robustness. Based on this reviewer’s comment, we describe this methodological approach and associated limitations in the Strengths and Limitations section of our Discussion.

Because this paper limited the literature review to English journals, by necessity it missed literature that may exist on this subject from regions such as Africa and the Indian subcontinent where we see significant numbers of new journals emerging. While all studies need to be scoped for a variety of reasons, I would have liked to have seen the authors make more of a nod to that limitation, rather than just noting it as a limitation of the language.

• We have acknowledged the increase and growth of journals in non-English speaking countries and the potential findings that we may have missed in the limitations section of our manuscript.

Another thing worth noting is that some of the descriptions of the operational modalities of predatory journals may actually be instructive in constructing guidelines in this case. While such journals are not a healthy mechanism for scientific discovery, they have been successful in recruiting unsuspecting new researchers and in publicizing themselves. A future direction for this research may involve studying the tools and methods those journals employ to see if those methodologies could be turned to more constructive purposes. Such a direction may be worth a nod in the conclusion.

• We have included a few sentences regarding this future research interest within the Future Directions section.

I do not see sources 7, 16, 17, 18, 19, 20, 21, or 35 noted in the body of the text.

• Thank you for catching this. This has now been corrected.

In addition to the data included in the appendices the authors should consider uploading RIS (or a similar format) files containing raw and distilled literature search results to OSF.

• We have uploaded the RIS file to OSF. It can now be found here (in the “Raw Data” folder): https://osf.io/6q5ua/files/osfstorage

With those minor points made I feel like this is a high-quality paper that covers an important subject using a sound methodology. I recommend it for publication with minor revisions.

Reviewer #3: The idea of this paper is both somewhat interesting and relevant, albeit probably only to a limited group of professional: To identify and describe any existing guidelines for starting a scholarly journal. And to investigate whether this can be done through a scoping review methodology.

• We kindly thank this reviewer for providing their feedback on our manuscript.

1) The title promises a scope related to creating scholarly journals as such but only delivers on biomedical journals. Perhaps it could be emphasized in the subtitle that this is indeed the point of the scoping review?

• We agree and have changed the title to focus on biomedical journals.

2) Sources: It was curious to see learn that the authors have chosen a host of classical medical science databases for this study, since this would naturally limit identified sources to those fields, while other sources external to medicine might also contain relevant information. The authors could add a little more as to the reasoning behind the choice of primary databases and what this means to potential biases. Also, the authors could spend a few lines arguing why they have not included libraries as a source of relevant information, or potentially information from publishers internal sources.

• While many of our sources were medical science databases, we should note that we also searched databases such as ERIC which focuses on education research, and Web of Science, which broadly includes all fields of science. We also included several grey literature sources which are external to the medical sciences. Moreover, an academic librarian was consulted to assist us in choosing our resources to search, and the database search strategy was peer reviewed (PRESSed) by an additional academic librarian. Overall, these sources were recommended based on the advice of two academic librarians, to best balance between comprehensiveness and project feasibility. We acknowledge that not including libraries may still reflect a limitation of our study, and mention of this has been added to the Strengths and Limitations section of our Discussion.

3) Method: Giving the comments above, it is my impression that the authors put too much emphasis on their methodology when discussing their points and in reaching conclusions. Statistically they are dealing with very low numbers and sources of limited riceness. While the scoping review has been carried out well it seems, its use in discussing the topic is clearly limited.

• We agree that there are limitations to this review and we have addressed these in the Strengths and Limitations section of our Discussion. 

4) Results: It is hardly surprising that very few sources are identified (33), but perhaps striking that more than 5000 unique sources are obtained initially. Given the subject field and databases etc. It is to be expected that relevant titles are not abundant.

• While it may not be surprising that we identified very few sources, we would argue that the importance of this scoping review lies not in the fact that we found a small number of courses, but rather that we identified what were the sources that provide recommendations for the creation of a new journal. We mentioned in our manuscript that we were interested in research related recommendations for creating new journals, rather than business related ones. We believe that the latter would be more likely to be found in the publisher's internal guidance, whereas the research related recommendations would likely be found in the academic literature. 

5) Conclusions: I would recommend that the authors consider the possible effects on their conclusions of not having access to publishers' internal information on starting journals. This would probably require a more qualitative approach, however.

• We acknowledge this reviewer’s point, and while this was beyond the scope of this review, we have noted in our manuscript that future research could involve contacting different publishers and requesting that they share their internal information for analysis.

6. PLOS authors have the option to publish the peer review history of their article (what does this mean?). If published, this will include your full peer review and any attached files.

Do you want your identity to be public for this peer review? For information about this choice, including consent withdrawal, please see our Privacy Policy.

Reviewer #1: Yes: Pragya Lodha

Reviewer #2: Yes: Jason Bengtson

Reviewer #3: No

Should your editorial office require any further edits following my most recently submitted submission, please do not hesitate to inform me and we will make these changes as soon as possible. Thank you for your consideration.

Yours sincerely,

Dr. Jeremy Y. Ng, MSc, PhD

Centre for Journalology, Ottawa Methods Centre, Ottawa Hospital Research Institute

---

## [Decision Letter · Decision Letter 1]

9 Feb 2023

Recommendations and Guidelines for Creating Scholarly Biomedical Journals: A Scoping Review

PONE-D-22-21604R1

Dear Dr. Jeremy Y. Ng,

We’re pleased to inform you that your manuscript has been judged scientifically suitable for publication and will be formally accepted for publication once it meets all outstanding technical requirements.

Kind regards,

Shahabedin Rahmatizadeh, Ph.D.

Academic Editor

PLOS ONE

Reviewers' comments:

Reviewer's Responses to Questions

**Comments to the Author**

1. If the authors have adequately addressed your comments raised in a previous round of review and you feel that this manuscript is now acceptable for publication, you may indicate that here to bypass the “Comments to the Author” section, enter your conflict of interest statement in the “Confidential to Editor” section, and submit your "Accept" recommendation.

Reviewer #1: All comments have been addressed

Reviewer #2: All comments have been addressed

Reviewer #3: All comments have been addressed

2. Is the manuscript technically sound, and do the data support the conclusions?

Reviewer #1: Yes

Reviewer #2: Yes

Reviewer #3: Yes

3. Has the statistical analysis been performed appropriately and rigorously? 

Reviewer #1: N/A

Reviewer #2: Yes

Reviewer #3: N/A

4. Have the authors made all data underlying the findings in their manuscript fully available?

Reviewer #1: Yes

Reviewer #2: Yes

Reviewer #3: No

5. Is the manuscript presented in an intelligible fashion and written in standard English?

Reviewer #1: Yes

Reviewer #2: Yes

Reviewer #3: Yes

6. Review Comments to the Author

Reviewer #1: The authors have satisfactorily addressed all the comments, and I have no further comments. The manuscript looks promising.

Reviewer #2: I appreciate the authors' attention to reviewer feedback and I believe this paper is ready for acceptance.

Reviewer #3: Given the authors' responds to the comments to my report, I recommend that the revised version of the manuscript is published.

7. PLOS authors have the option to publish the peer review history of their article (what does this mean?). If published, this will include your full peer review and any attached files.

Reviewer #1: **Yes: **Pragya Lodha

Reviewer #2: **Yes: **Jason A Bengtson

Reviewer #3: **Yes: **Bertil Fabricius Dorch

---

## [Editor Report · Acceptance letter]

17 Feb 2023

PONE-D-22-21604R1 

Recommendations and Guidelines for Creating Scholarly Biomedical Journals: A Scoping Review 

Dear Dr. Ng:

I'm pleased to inform you that your manuscript has been deemed suitable for publication in PLOS ONE. Congratulations! Your manuscript is now with our production department. 

Kind regards, 

on behalf of

Dr. Shahabedin Rahmatizadeh 

Academic Editor

PLOS ONE